# Vitamin D Status, CMV Seropositivity, and Viral Cytokine Expression in Pregnancy

**DOI:** 10.3390/v17091203

**Published:** 2025-08-31

**Authors:** Adalvan D. Martins, Jennifer Woo, Brandi Falley, Juliet V. Spencer

**Affiliations:** 1Division of Biology, Texas Woman’s University, Denton, TX 76204, USA; amartins@twu.edu; 2College of Nursing and Health Innovation, University of Texas at Arlington, Arlington, TX 76019, USA; jennifer.woo@uta.edu; 3Division of Mathematics, Texas Woman’s University, Denton, TX 76204, USA; bfalley@twu.edu

**Keywords:** CMV, cmvIL–10, congenital infection, cytokine, cytomegalovirus, vitamin D, maternal health, pregnancy

## Abstract

Cytomegalovirus (CMV) is the leading infectious cause of birth defects and has been linked to increased risk of preterm birth (PTB). CMV establishes lifelong latency and is more prevalent among Black and Hispanic/Latina women, populations already at higher risk for adverse pregnancy outcomes. Vitamin D deficiency, also common in these groups, has been linked to impaired immune function and increased susceptibility to infections, including CMV. In this cross–sectional study of 63 pregnant minority women (50 CMV+, 13 CMV−), we evaluated associations among serum 25(OH)D levels, CMV serostatus, and cmvIL–10, the CMV–encoded interleukin–10 homolog that modulates host immune responses. While vitamin D insufficiency and CMV seropositivity were both highly prevalent, we found no statistically significant associations between 25(OH)D levels and CMV serostatus or cmvIL–10 levels. These findings highlight the need for further investigation into how vitamin D deficiency and CMV infection may independently or synergistically contribute to maternal and neonatal health disparities.

## 1. Introduction

Human cytomegalovirus (CMV), a ubiquitous member of the *Herpesviridae* family, is the most common congenital infection worldwide. In pregnancy, both primary CMV infection and reactivation of latent CMV can lead to vertical transmission, a serious threat to fetal health. Nearly 30,000 infants are born with congenital CMV each year in the United States, about 1 in 150 births, and 20% of these result in severe fetal outcomes, including microcephaly, sensorineural hearing loss, and neurodevelopmental delays [1]. Congenital CMV causes more long–term disability than more widely recognized conditions such as Down syndrome, fetal alcohol syndrome, or spina bifida [2]. CMV is a major, under–recognized public health problem, and emerging research has also implicated CMV infection as a contributing factor in preterm birth (PTB). CMV may disrupt pregnancy placental inflammation, impair trophoblast invasion, and cause immune dysregulation at the maternal–fetal interface [3,4,5] highlighting the need to better understand how CMV impacts pregnancy and contributes to PTB.

PTB, defined as delivery before 37 weeks of gestation, remains a significant contributor to neonatal morbidity and mortality. Both the incidence and complications of PTB are elevated in minority populations [6]. Thus, identifying modifiable biological risk factors is a critical priority. Among such factors, vitamin D deficiency has gained attention not only for its classical role in bone metabolism but also for its increasingly recognized function in immune regulation and infection control during pregnancy [7,8].

25–hydroxyvitamin D [25(OH)D] (hereafter referred to as vitamin D) is essential for modulating innate and adaptive immunity, partly through the upregulation of antimicrobial peptides such as cathelicidins and β–defensins, which are crucial for controlling infections. Deficiency of vitamin D is highly prevalent among pregnant women, especially in individuals with darker skin pigmentation, limited sun exposure, or inadequate dietary intake, factors more common in Black and Hispanic/Latina populations [9]. Low maternal vitamin D levels have been independently associated with a spectrum of adverse outcomes, including PTB, preeclampsia, and fetal growth restriction [10,11,12].

Recent literature suggests that vitamin D status may also play a role in modulating CMV infection and its outcomes. Experimental studies have shown that CMV can downregulate the vitamin D receptor (VDR) in infected cells, potentially impairing downstream immunomodulatory effects of vitamin D [13]. Clinical reports further describe cases where vitamin D deficiency has been associated with enhanced CMV pathogenicity, including CMV–induced enteritis and vascular damage [14,15]. Vitamin D supplementation has also been shown to reduce herpesvirus reactivation risk, including CMV, in immunocompromised and pregnant populations [16].

A unique feature of CMV pathogenesis is the virus’s ability to evade host immune defenses through the production of cmvIL–10, the CMV–encoded interleukin–10 homolog. This viral cytokine suppresses host pro–inflammatory responses and antigen presentation, promoting viral persistence [17,18]. In the context of vitamin D deficiency, the effects of cmvIL–10 may be amplified, increasing maternal susceptibility to CMV reactivation and associated complications like PTB.

Despite growing interest in the independent effects of vitamin D deficiency and CMV infection on pregnancy, few studies have examined their combined influence. The convergence of these risk factors is particularly relevant in minority populations, where both vitamin D deficiency and CMV seroprevalence are disproportionately high [19]. These dual exposures may synergistically impair placental immunity and increase the risk of PTB.

Based on these considerations, we investigated whether maternal CMV serostatus was associated with vitamin D levels in a cohort of predominantly Black and Latina pregnant women who are at elevated risk for preterm birth. This analysis explored the potential intersection of viral infection and nutrient status in a population disproportionately affected by both CMV and vitamin D deficiency. CMV is known to employ multiple strategies to evade host immunity and disrupt pregnancy outcomes [20], underscoring the importance of examining viral immune modulators such as cmvIL–10 in this context. Our aim is to provide foundational data to inform future studies on how co–occurring exposures may contribute to immune dysregulation and adverse pregnancy outcomes.

## 2. Materials and Methods

### 2.1. Study Population and Sample Collection

Serum samples were collected from 63 pregnant women receiving care at two large urban obstetrics clinics in the Dallas/Fort Worth metroplex, a region with a predominantly Medicaid–insured population. Participants were between 24 and 32 weeks of gestation and self–identified as Black and Hispanic/Latina. Samples were analyzed for serum 25(OH) vitamin D, CMV and Epstein–Barr Virus (EBV) serostatus, and the viral cytokine cmvIL–10. Informed consent was obtained from all participants in accordance with an approved Institutional Review Board protocol, and participants in this cohort consented to biobanking of their samples for future studies including the current study (IRB–FY2021–286). 

### 2.2. Vitamin D Levels

Serum 25(OH)D levels, the primary circulating form of vitamin D, were measured using Liquid Chromatography–Tandem Mass Spectrometry (LC–MS/MS) standard of care clinical assay performed by Quest Diagnostics (Denton, TX, USA). 

### 2.3. CMV and EBV Serostatus

CMV IgG and IgM serostatus was assessed using commercial test kits Okay (Arlington Scientific, Springville, UT, USA), following the manufacturer’s protocol. CMV serostatus was determined based on the quantification of IgG and IgM antibodies, represented by the Immune Status Ratio (ISR). ISR was calculated using the ratio of optical density readings from the sample and a calibrator. EBV IgG status was also determined by using a commercial test kit according to the manufacturer’s instructions (Arlington Scientific).

### 2.4. cmvIL–10 Quantification

Levels of cmvIL–10 were measured via enzyme–linked immunosorbent assay (ELISA), as described [21]. Briefly, ELISA plates were coated with capture antibodies directed to cmvIL–10 (cmvIL–10 affinity purified Polyclonal Ab, Goat IgG, R&D Systems, Minneapolis, MN, USA) and incubated with diluted serum samples, in triplicate. A secondary antibody directed to cmvIL–10 was added (Viral HCMV IL–10 Biotinylated Affinity Purified Polyclonal Ab, Goat IgG, R&D Systems), followed by streptavidin conjugated with horseradish peroxidase (SA–HRP). Substrate solution was added for colorimetric detection. Absorbance was measured at 450 nm on a Synergy H plate reader (Biotek, Winooski, VT, USA) and cytokine concentrations interpolated from a standard curve of recombinant cmvIL–10 protein (R&D Systems).

### 2.5. Statistical Analysis

Descriptive statistics were used to analyze the study data. Given the small sample sizes and non–normal distribution observed in some groups, non–parametric methods were used for group comparisons. The Kruskal–Wallis test was applied to assess differences across multiple groups, and the Mann–Whitney U test was used for pairwise comparisons between two independent groups. Statistical significance was determined using a two–tailed *p*-value threshold of 0.05. All statistical analyses and graph generation were performed using GraphPad Prism (version 8.4.3, GraphPad Software, San Diego, CA, USA).

## 3. Results

### 3.1. Study Population

The study included 63 pregnant individuals (37 Hispanic/Latina, 26 Black), with a mean maternal age of 30.57 ± 5.63 years and a mean gestational age of 29.66 ± 2.99 weeks at recruitment and blood draw. Participants had an average of 3.02 ± 1.7 total pregnancies, with a median of one full–term pregnancy and an average of 0.24 ± 0.56 PTBs and 0.70 ± 1.01 miscarriages. Detailed characteristics of the study population are shown in Table 1.

### 3.2. Serological Status

To determine which participants harbored CMV, we tested all serum samples for CMV–specific IgG antibodies, which indicate prior infection. The results showed that 79.4% (*n* = 50) of participants were CMV IgG + (Table 1). To assess potential recent or active infection, we measured CMV–specific IgM antibodies. None of the samples tested positive for CMV IgM. We also tested for EBV IgG, as EBV is a related herpesvirus with similar patterns of latency and reactivation. We found that 95.2% of the participants were positive for EBV IgG. 

### 3.3. Vitamin D and Ethnicity

Serum 25(OH)D levels were categorized into three groups: sufficient (≥30 ng/mL, *n* = 24), insufficient (20–29 ng/mL, *n* = 23), and deficient (<20 ng/mL, *n* = 16), as defined by the Endocrine Society [22]. The mean serum 25(OH)D levels differed significantly across these groups (*p* < 0.0001), with average values of 38.75 ng/mL in the sufficient group, 24.43 ng/mL in the insufficient group, and 13.50 ng/mL in the deficient group (Figure 1a). Overall, 62% of participants (*n* = 39) had insufficient or deficient levels of vitamin D. We then asked whether vitamin D status differed between Black and Hispanic/Latina participants, but no statistically significant differences were observed between racial/ethnic groups (Figure 1b). However, a higher proportion of Black participants had vitamin D deficiency (34.6%) compared to Hispanic participants (18.9%), suggesting a potential trend that may warrant further investigation in a larger cohort.

### 3.4. Vitamin D, Serostatus and IgG Levels

Given the known role of vitamin D in supporting immune function, we next asked whether CMV seropositivity might be associated with lower vitamin D levels. However, when we compared mean vitamin D concentrations between CMV+ and CMV− participants, there were no significant differences, suggesting that prior CMV infection does not appear to contribute to vitamin D deficiency in this cohort (Figure 2a). To explore this relationship further, we assessed whether the intensity of the immune response, as measured by ISR for CMV and EBV IgG, varied with vitamin D status. High ISR values can reflect heightened antibody production, which may occur during infection or chronic immune activation. No significant differences in ISR were observed across vitamin D categories for either CMV+ or EBV+ participants (*p* = 0.22029 and *p* = 0.522, respectively; Figure 2b,d). We also examined whether these patterns differed by racial/ethnic groups. Although average ISR values for CMV IgG were higher among Black women with insufficient or deficient vitamin D compared to their Hispanic/Latina counterparts, this trend did not reach statistical significance (Figure 2c). No significant differences were observed in EBV ISR values (Figure 2e). It is possible that with a larger sample size, subtle differences in CMV immune activation related to both vitamin D status and racial/ethnic groups may become more apparent.

### 3.5. cmvIL–10 Versus Vitamin D Levels and Ethnicity

To explore whether vitamin D status might influence viral immune modulation, we evaluated levels of the viral cytokine cmvIL–10 across the three vitamin D categories in CMV+ participants (Figure 3a). Interestingly, the group with insufficient vitamin D had the highest average cmvIL–10 concentration (77.08 pg/µL), compared to 36.26 pg/µL in the deficient group and 18.88 pg/µL in the sufficient group. Despite this trend, the differences were not statistically significant (*p* = 0.560), likely due to considerable variability in cmvIL–10 levels in the insufficient group. These findings suggest that, within this cohort, vitamin D status does not have a consistent or predictable effect on cmvIL–10 expression. Similarly, when we compared cmvIL–10 levels between Hispanic/Latina and Black participants within each vitamin D category, no differences were observed, indicating that racial/ethnic background did not appear to influence cmvIL–10 production in this cohort.

## 4. Discussion

This study explored the relationship between maternal vitamin D levels and CMV serostatus in pregnant minority women at risk of PTB. Seroprevalence rates for both CMV (79.4%) and EBV (95.2%) were high in the study group relative to the national averages reported in the general population of the United States [23,24]. This elevated prevalence is consistent with prior findings that CMV seropositivity is significantly higher among socioeconomically disadvantaged and minority populations, particularly Black and Hispanic/Latina women [25]. The absence of CMV IgM reactivity indicates no active or recent infection among participants. Notably, cmvIL–10 levels showed considerable variability, with a few individuals displaying markedly elevated levels. This variation may reflect differences in immune regulation or the possibility of subclinical viral reactivation in CMV+ individuals [18].

In this cohort, average maternal vitamin D levels were 27.11 ng/mL, with 62% of participants falling into the insufficient or deficient range; as defined by the Endocrine Society [22]. This is concerning as vitamin D deficiency is associated with adverse pregnancy outcomes; including PTB, and supplementation is recommended for pregnant women with a vitamin D level <40 ng/mL [26]. Although Vitamin D deficiency has been associated with PTB [27], the precise mechanisms remain multifactorial and incompletely understood. Emerging evidence suggests that vitamin D influences placental development; immune tolerance, and inflammatory modulation during gestation. Low levels of maternal vitamin D have been linked to increased levels of pro–inflammatory cytokines, which can disrupt the maternal–fetal interface and potentially trigger preterm labor [28].

### 4.1. Ethnic Disparities in Vitamin D Status and CMV Exposure

This study reaffirmed that Black and Hispanic/Latina women have a high prevalence of vitamin D insufficiency and CMV seropositivity, which aligns with nationwide data indicating greater CMV exposure and lower vitamin D levels in minority populations [29,30,31]. Although the difference in vitamin D deficiency between Black and Hispanic/Latina participants was not statistically significant in this cohort, previous findings have shown that skin pigmentation, dietary patterns, and reduced sun exposure contribute significantly to these disparities [9,32]. In addition, vitamin D deficiency can broadly increase susceptibility to herpesvirus infections by shifting the balance of pro– and anti–inflammatory cytokines, reinforcing the importance of clarifying its role in CMV immune control [33].

### 4.2. Lack of Correlation Between CMV Serostatus, cmvIL–10, and 25(OH)D Levels

Although prior literature suggests that vitamin D enhances the host’s antiviral immunity, particularly through upregulation of cathelicidin and β–defensin expression, which are critical in early viral containment [34], our results did not indicate any meaningful difference in vitamin D levels between CMV+ and CMV− pregnant women in the early to late third trimester of pregnancy. This could be due to the lack of patients with an active CMV infection, as latent infections may not significantly affect immune parameters, and possibly due to sample size limitations and timing of blood draw. Our data showed no significant variation in cmvIL–10 levels across all different 25(OH)D levels. cmvIL–10 mimics host IL–10 to suppress immune responses and facilitate viral persistence [21], and their correlation with 25(OH)D remains poorly understood [35,36]. Recent transcript profiling further demonstrates that both cmvIL–10 and LacmvIL–10 are expressed not only during productive infection but also at lower levels in latent infection. Importantly, in renal transplant recipients, cmvIL–10 transcripts were consistently detected up to 60 days post–transplantation and often preceded detectable viral DNA, underscoring its potential as an early biomarker of reactivation [37].

One interesting finding in this research was the presence of high levels of cmvIL–10 in two CMV–seronegative pregnant women. Other studies have also detected cmvIL–10 in CMV− samples, which are generally attributed to subclinical infection that did not result in seroconversion [21,38]. This finding highlights the potential of cmvIL–10 to subvert the host’s protective immune response to CMV, potentially inhibiting effective seroconversion [39]. Since cmvIL–10 can compete with human IL–10 for binding sites, and vitamin D supplementation can lead to increased serum concentrations of IL–10 [40], larger–scale studies are necessary to better understand their relationship.

### 4.3. EBV Status, Vitamin D, and Ethnicity

In our study, EBV IgG seropositivity was nearly universal (95.2%), with ISR levels showing no statistically significant difference between Black and Hispanic/Latina participants. This contrasts with some prior research, especially in non–pregnant cohorts, where racial disparities in EBV serostatus were observed [41]. The absence of such differences here may be due to our sample’s relatively small size or more homogeneous environmental exposures and stressors across the groups. 

Vitamin D deficiency, as evidenced by our cohort’s mean level of 27.11 ng/mL, may play a central role in EBV immune dynamics. Elevated levels of EBV IgG were observed in this study. Multiple sclerosis studies have shown that low vitamin D correlates with heightened EBV reactivity, and that supplementation can reduce antibodies against Epstein–Barr nuclear antigen–1 (EBNA-1), the key viral latency antigen [42]. In pregnancy, where immune control of latent infections is critical, vitamin D deficiency might similarly permit increased EBV reactivation, reflected in elevated IgG ISR across the samples. Thus, even in the absence of ethnic differences, vitamin D insufficiency could be a crucial shared risk factor for altered EBV immune control in this population.

### 4.4. Vitamin D, Cytomegalovirus, and PTB

Several large–scale cohort studies have associated maternal vitamin D deficiency in early pregnancy with an increased risk of PTB, suggesting a possible window of vulnerability during placentation and immune tolerance establishment [10,43]. More recently, growing attention has been directed toward the role of cytomegalovirus (CMV) as a latent or reactivating infection that may contribute to PTB via immune dysregulation, placental inflammation, and vertical transmission, associated with vitamin D deficiency [44,45,46]. Interestingly, vitamin D may play a protective role against such CMV–mediated inflammation. In vitro and in vivo studies have shown that vitamin D enhances innate immunity against viral infections, and its deficiency may compromise antiviral responses and facilitate herpesvirus reactivation, including CMV [33]. These findings suggest a synergistic pathogenic interaction between hypovitaminosis D and latent viral activation.

Moreover, cmvIL–10, a viral cytokine encoded by CMV that helps evade host immunity, plays a significant role in suppressing maternal immune responses. While our study did not detect statistically significant differences in cmvIL–10 levels across vitamin D strata, its persistent presence in CMV–seropositive women supports its role in immune modulation. This may reflect a compensatory immune evasion strategy by CMV, which is potentiated in environments with compromised immune surveillance, such as vitamin D deficiency.

## 5. Conclusions

In summary, this study provides a comprehensive analysis of the interplay between vitamin D status and CMV serostatus in a cohort of pregnant minority women, a population disproportionately affected by both vitamin D deficiency and CMV exposure. In our relatively small sample size of 63 pregnant women, no statistically significant associations were found among 25(OH)D levels, CMV serostatus, and cmvIL–10 levels. The potential immunomodulatory role of cmvIL–10, even in seronegative individuals, and the possibility of subtle effects of vitamin D on latent viral activity highlight complex, time–sensitive interactions that may not be captured in a cross–sectional design.

Although our study focused on CMV serostatus rather than distinguishing primary from recurrent infection, the absence of CMV IgM suggests that most CMV+ participants harbored latent infections. In high–seroprevalence populations, reactivation or reinfection during pregnancy is more common than primary infection and can still result in congenital transmission and adverse outcomes, even if transmission rates are lower than with primary infection. Immune modulation during pregnancy, particularly in the setting of micronutrient deficiencies such as vitamin D, may increase susceptibility to viral reactivation [16,44]. Future longitudinal studies incorporating CMV DNA detection, IgG avidity testing, and serial measurements of viral immune modulators like cmvIL–10 will be critical to understanding how vitamin D status influences reactivation risk and outcomes, and to identifying intervention points in high–prevalence, high–risk populations. Additionally, future studies with larger sample sizes and trimester–specific sampling would help clarify how vitamin D status influences viral immune responses during pregnancy and identify potential factors that will help reduce maternal–fetal health outcomes.

## Figures and Tables

**Figure 1 viruses-17-01203-f001:**
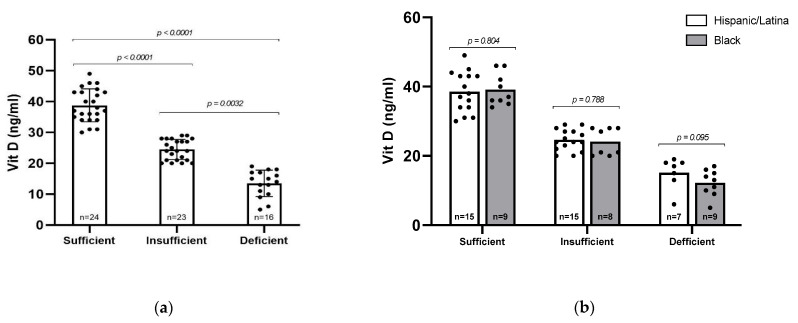
Vitamin D (25–OH) levels among participants. (**a**) All participants were sorted by Vitamin D sufficiency level and analyzed via Kruskal–Wallis test. (**b**) All participants were sorted by sufficiency level and racial/ethnic group. Statistical comparisons were performed via Mann–Whitney U test, with *p*-values indicated. Error bars represent mean ± standard deviation; individual data points are shown.

**Figure 2 viruses-17-01203-f002:**
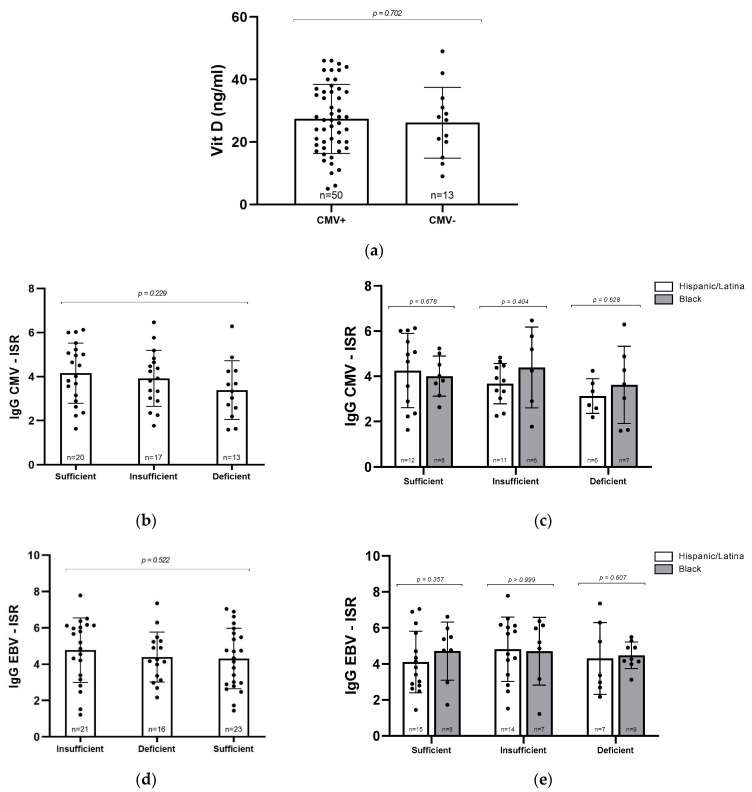
Relationship between vitamin D levels and CMV and EBV serostatus. (**a**) Vitamin D concentrations (ng/mL) in CMV+ (*n* = 50) and CMV− (*n* = 13) samples. (**b**) CMV IgG ISR levels in participants grouped by vitamin D status. (**c**) CMV IgG ISR levels by vitamin D level and racial/ethnic group. (**d**) EBV IgG ISR levels by vitamin D status. (**e**) EBV IgG ISR levels by vitamin D level and racial/ethnic group. Statistical analyses were conducted via Mann–Whitney U test (**a**,**c**,**e**) and Kruskal–Wallis test (**b**,**d**). Error bars represent mean ± standard deviation; individual data points are shown.

**Figure 3 viruses-17-01203-f003:**
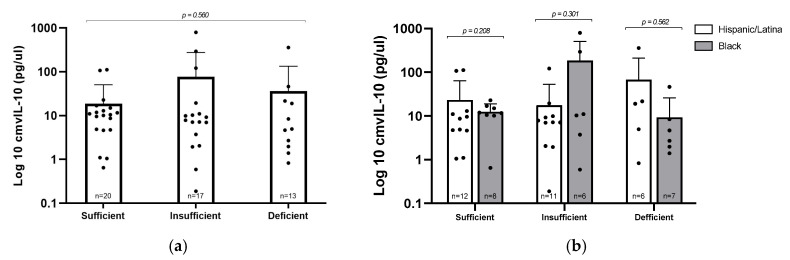
cmvIL–10 levels by vitamin D status and racial/ethnic group. (**a**) Log_10_–transformed concentrations of cmvIL–10 were compared by vitamin D levels; statistical analysis via Kruskal–Wallis test. (**b**) cmvIL–10 levels stratified by vitamin D status and racial/ethnic group; statistical analysis via Mann–Whitney U test. Error bars represent mean ± standard deviation of log–transformed values; individual data points are plotted.

**Table 1 viruses-17-01203-t001:** Characteristics of the Study Population.

***n* = 63**	**Mean (SD)**	**Median**	**Range**
Maternal Age (years)	30.57 (5.63)	31	19–43
Gestational Age (weeks)	29.66 (2.99)	30.3	24–34.3
Gravida (total pregnancies)	3.02 (1.70)	3	1–8
Full Term Pregnancies	1.13 (1.20)	1	0–5
Preterm Pregnancies	0.24 (0.56)	0	0–3
Miscarriages Pregnancies	0.70 (1.01)	0	0–4
Number of Living Children	1.37 (1.35)	1	0–5
Vitamin D, 25–OH Levels (ng/mL)	27.11 (11.03)	27	5–49
Sufficient (≥30 ng/mL)	38.75 (5.33)	37.5	30–49
Insufficient (20–29 ng/mL)	24.43 (3.29)	24.00	20–29
Deficient (<20 ng/mL)	13.50 (4.27)	14.50	5–19
cmvIL–10 (pg/mL)	65.09 (182.90)	8.69	0–879.88
** *n* ** ** = ** **63**	**% Positive**	**% Negative**	**Mean ISR (SD)**
Cytomegalovirus (CMV) IgG	79.4 (*n* = 50)	20.6 (*n* = 13)	3.88 ± 1.33
Cytomegalovirus (CMV) IgM	0	100	-
Epstein–Barr Virus (EBV) IgG	95.2 (*n* = 60)	1.6 (*n* = 1) *	4.54 ± 1.65

* 2 participants had EBV ISR values in the equivocal range (0.9–1.1) even with multiple re–tests. SD = standard deviation.

## Data Availability

The data presented in this study are available upon reasonable request from the corresponding author. The data are not publicly available due to privacy concerns and institutional policies.

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
