# Peer review of "Vitamin D Status, CMV Seropositivity, and Viral Cytokine Expression in Pregnancy"

_viruses, 2025, doi:10.3390/v17091203_

Round 1

Reviewer 1 Report

Comments and Suggestions for Authors

Martins and co-authors present the role of vitamin D relative to CMV infection in a cross-sectional study of pregnant women. As CMV is a serious complication for pregnancy outcomes (as is vitamin D deficiency) understanding the connection (if any) and interplay of these factors is key to understanding pre-term birth and birth defects in humans. The authors compared CMV serostatus (seropositivity, ISR, and IgG vs IgM conversion) with vitamin D levels across their patient cohort. The authors found some interesting trends including that their cohort represents a solid cross-section of vitamin D levels in the population; and that in this cohort CMV status at the broad level is not correlative. Interestingly, they did find that preliminary information that the CMV homolog to IL-10 (a key immune modulator) may be altered and correlate with vitamin D status. This finding has implications for the interconnectedness of immune activation with vitamin D deficiency. Overall, this a solid, straightforward study that cleanly addresses several biological questions with direct relevance to human health. I have only a few minor comments for the authors.

Minor comments:

  • Parameters of how LC-MS/MS was performed (or if this is a standard of care clinical assay) would be useful in the methods.
  • Information on the EBV serostatus testing (results 3.2). I do not see that the method is included in the methods section.
  • Are the values for vitamin D deficiency vs insufficiency (results section 3.3) standard values? References? If not, how were these determined as the cutoff values? Defining this would strengthen the claims in the discussion (starting line 221) that this study reaffirms these populations have a high prevalence of vitamin D insufficiency by allowing comparison to historical data (as other populations were not directly compared in this study).
  • It would be interesting if the authors could briefly discuss some of the other data on latent infection/reactivation of established CMV infections vs primary infection in pregnancy, especially with regards to their future directions/conclusions section.
  • There are some stray typographical errors/changes (e.g., random double spaces, a couple of typos) the authors may want to fix prior to the proof stage to avoid later issues.

Author Response

Author Response : 

Thank you very much for taking the time to review this manuscript. Please find the detailed responses below and the corresponding revisions indicated with track changes in the re-submitted files.

Questions for General Evaluation:

Response: the reviewer indicated “Yes” for each question except for “Are the methods adequately described?”, for which the response was can be improved.  The methods section was edited to include additional information, as requested below.

Minor comments:

1. Parameters of how LC-MS/MS was performed (or if this is a standard of care clinical assay) would be useful in the methods.

Response: We thank the reviewer for this comment. We have clarified this in the Methods section that the LC-MS/MS was a standard of care clinical assay performed by Quest Diagnostics (line 95).

2. Information on the EBV serostatus testing (results 3.2). I do not see that the method is included in the methods section.

Response: We regret that this information was not clearly articulated.  We have updated the Methods section to include more details about EBV testing (lines 101-103).

3. Are the values for vitamin D deficiency vs insufficiency (results section 3.3) standard values? References? If not, how were these determined as the cutoff values? Defining this would strengthen the claims in the discussion (starting line 221) that this study reaffirms these populations have a high prevalence of vitamin D insufficiency by allowing comparison to historical data (as other populations were not directly compared in this study).

Response: We thank the reviewer for pointing out this oversight.  We have added the clarification that these are Endocrine Society guidelines with the appropriate reference. (Lines 144-145).

4. It would be interesting if the authors could briefly discuss some of the other data on latent infection/reactivation of established CMV infections vs primary infection in pregnancy, especially with regards to their future directions/conclusions section.

Response: We appreciate this feedback. We have added a paragraph to the conclusions to weave in the concept of primary infection vs latent infection and reactivation to our discussion of CMV in pregnant women.

5. There are some stray typographical errors/changes (e.g., random double spaces, a couple of typos) the authors may want to fix prior to the proof stage to avoid later issues.

Response:  We thank the reviewer for careful reading. We have carefully reviewed and edited the final manuscript to address minor typos.

Reviewer 2 Report

Comments and Suggestions for Authors

This manuscript examined the association between Vitamin D status and CMV-driven immunomodulation in 63 Black and Hispanic/Latina pregnant women. Results revealed no significant link between Vitamin D levels and either CMV infection or cmvIL-10 expression. Nonetheless, the findings underscore the “double jeopardy” of Vitamin D deficiency and CMV infection confronting this maternal population. However, the article also has the following shortcomings:

Major problems:

  1. This study was hampered by an underpowered sample: only 63 pregnant women were enrolled, and the marked imbalance across subgroups likely obscured genuine between-group differences.
  2. Using CMV-specific IgM seronegativity as the sole criterion for ruling out recent or active infection may overlook cases of low-level replication or viral re-activation. Incorporating CMV DNA quantification would provide a more definitive exclusion of active infection.
  3. The biological significance of the cmvIL-10 findings remains ambiguous. Notably, the Vitamin D deficient group exhibited the highest mean cmvIL-10 concentrations—an unexpected pattern that warrants targeted mechanistic investigation and clearer contextualization.

Minor problems:

  1. Terminological consistency needs tightening.
  2. It is recommended that the manuscript incorporate the most recent literature to enhance its timeliness and scientific rigor.

Comments on the Quality of English Language

The English expression is generally accurate; however, some terminology needs to be standardized.

Author Response

Author Response : 

Questions for General Evaluation:

Response:  The reviewer indicated “Can be improved” for each question, and we provide clarification on the specific improvements in our point by point response below.  Other improvements were made as a result of Reviewer 1’s comments, as indicted above.

Major Problems:

1.This study was hampered by an underpowered sample: only 63 pregnant women were enrolled, and the marked imbalance across subgroups likely obscured genuine between-group differences.

Response:  We thank the reviewer for this feedback and agree that the study would benefit from additional enrollment.  Regrettably, we could not recruit additional women and believe that publishing the findings from the current study may help future researchers who are able to achieve more robust enrollment.

2.Using CMV-specific IgM seronegativity as the sole criterion for ruling out recent or active infection may overlook cases of low-level replication or viral re-activation. Incorporating CMV DNA quantification would provide a more definitive exclusion of active infection.

Response:  We appreciate this feedback and agree that DNA quantification would have been useful.  Unfortunately, we have already freeze-thawed the samples multiple times and any attempt to detect viral DNA would be futile, based on extensive previous experience screening human blood samples for indications of CMV infection.

3.The biological significance of the cmvIL-10 findings remains ambiguous. Notably, the Vitamin D deficient group exhibited the highest mean cmvIL-10 concentrations—an unexpected pattern that warrants targeted mechanistic investigation and clearer contextualization.

Response:  We thank the reviewer for this thoughtful comment and agree that the interpretation of the cmvIL-10 data is not straightforward. To clarify, while the Vitamin D deficient group exhibited higher mean levels of cmvIL-10, these differences were not statistically significant in our dataset. It remains possible that with a larger sample size, these differences could reach significance; however, we were limited by the number of samples available. We fully agree that the observed pattern raises interesting questions about underlying mechanisms, and we are actively pursuing studies aimed at elucidating the biological significance of cmvIL-10 in this context. 

Minor problems:

1. Terminological consistency needs tightening.

Response:  We thank the reviewer for careful reading of the manuscript, and we have made changes that improve the consistency of terminology use.  In particular, these corrections were made: we used “cmvIL-10” (fixed one “cvmIL-10” and several “viral HCMV IL-10” mentions); used “CMV” (replaced “HCMV” unless citing a source that uses it); used “25(OH)D” for lab results, “vitamin D” for general biological roles. We believe these changes improve the readability of the manuscript.

2. It is recommended that the manuscript incorporate the most recent literature to enhance its timeliness and scientific rigor.

Response:  We thank the reviewer for this valuable suggestion. In revision, we have incorporated several recent publications to strengthen the timeliness and scientific rigor of our manuscript.

  • Shang, Z.; Li, X. Human Cytomegalovirus: Pathogenesis, Prevention, and Treatment. Mol. Biomed. 2024, 5, 61. https://doi.org/10.1186/s43556-024-00226-7
    Added in the Introduction to frame our study within the most current understanding of CMV pathogenesis and immune evasion strategies, particularly those relevant to pregnancy outcomes.
  • Galdo-Torres, D.; Andreu, S.; Caballero, O.; Hernandez-Ruiz, I.; Ripa, I.; Bello-Morales, R.; Lopez-Guerrero, J.A. Immune Modulatory Effects of Vitamin D on Herpesvirus Infections. Int. J. Mol. Sci. 2025, 26(4), 1767. https://doi.org/10.3390/ijms26041767
    Added in the Discussion to highlight the latest evidence that vitamin D deficiency influences herpesvirus susceptibility by altering cytokine balance, reinforcing the rationale for exploring its role in CMV immune control.
  • Almeida, G.W.C.; et al. Expression Profile of Human Cytomegalovirus UL111A Transcripts (cmvIL-10 and LAcmvIL-10) in Productive and Latent Infection In Vivo and In Vitro. Viruses 2025, 17(4), 501. https://doi.org/10.3390/v17040501
    Added in the Discussion to provide new data on cmvIL-10 expression dynamics, showing that both cmvIL-10 and LAcmvIL-10 are detectable in latent infection and may precede detectable viral DNA, thereby supporting the biological relevance of our findings.

Together, these references update the manuscript with the most recent literature, strengthening both the rationale and interpretation of our results.

Round 2

Reviewer 2 Report

Comments and Suggestions for Authors I recommend accepting it in its current form, as the authors have answered all my questions.